# In Vitro or In Vivo Models, the Next Frontier for Unraveling Interactions between *Malassezia* spp. and Hosts. How Much Do We Know?

**DOI:** 10.3390/jof6030155

**Published:** 2020-08-28

**Authors:** Maritza Torres, Hans de Cock, Adriana Marcela Celis Ramírez

**Affiliations:** 1Grupo de Investigación Celular y Molecular de Microorganismos Patógenos (CeMoP), Departamento de Ciencias Biológicas, Universidad de los Andes, Carrera 1 N° 18A—12, Bogotá, Bogotá D.C. 11711, Colombia; marit-to@uniandes.edu.co; 2Microbiology, Department of Biology, Faculty of Science, Institute of Biomembranes, Utrecht University, Padualaan 8, 3584 CH Utrecht, The Netherlands; h.decock@uu.nl

**Keywords:** in vitro, in vivo, animal model, *Malassezia*, infection, host–pathogen interaction, *Galleria mellonella*

## Abstract

*Malassezia* is a lipid-dependent genus of yeasts known for being an important part of the skin mycobiota. These yeasts have been associated with the development of skin disorders and cataloged as a causal agent of systemic infections under specific conditions, making them opportunistic pathogens. Little is known about the host–microbe interactions of *Malassezia* spp., and unraveling this implies the implementation of infection models. In this mini review, we present different models that have been implemented in fungal infections studies with greater attention to *Malassezia* spp. infections. These models range from in vitro (cell cultures and ex vivo tissue), to in vivo (murine models, rabbits, guinea pigs, insects, nematodes, and amoebas). We additionally highlight the alternative models that reduce the use of mammals as model organisms, which have been gaining importance in the study of fungal host–microbe interactions. This is due to the fact that these systems have been shown to have reliable results, which correlate with those obtained from mammalian models. Examples of alternative models are *Caenorhabditis elegans*, *Drosophila melanogaster*, *Tenebrio molitor*, and *Galleria mellonella*. These are invertebrates that have been implemented in the study of *Malassezia* spp. infections in order to identify differences in virulence between *Malassezia* species.

## 1. Introduction

*Malassezia* is a lipid-dependent genus of yeasts found as commensals on human and animal skin [1,2]. Under specific conditions, these yeasts have been associated with skin diseases [3], Crohn’s disease, the exacerbation of colitis [4], Parkinson’s disease [5], pancreatic ductal adenocarcinoma [6], and fungemia [7,8,9] (Table 1). Factors determining the outcome of host–microbe interactions are multifactorial, involving environmental conditions like temperature and humidity, but also host factors and the predisposition of the host, which may be related to genetic factors and impairment in the immune response [10,11]. In addition, the virulence factors of *Malassezia* are likely to be involved. *Malassezia* spp. are generally regarded as opportunistic pathogens but how this skin commensal contributes to skin diseases remains a matter of debate. Studying the lifestyle of *Malassezia* spp. in model organisms is expected to contribute to unraveling this long-standing issue.

Even though *Malassezia* was described for the first time in 1846 and has been studied for a long time, relatively little is known about its interactions with the host. In part, this is due to the specific nutritional requirements of the yeast [1,2]. The fact that *Malassezia* requires fatty acids in media for growth has complicated the development of in vitro and in vivo models [12,13]. Many studies have addressed and compared the relative abundances of *Malassezia* species on the healthy and diseased skin of hosts [2]. Clearly, these species are regarded as skin commensals, which makes it more complex to determine their direct role in disease development. They were proposed to modulate the immune response through different mechanisms. For example, the composition of the cell wall contributes to the evasion of phagocytosis and a decrease in the release of proinflammatory cytokines by immune cells (IL-1β, IL-6 and TNF-α) [14], and the induction of IL-17 that leads to skin inflammation [15,16] and the indolic compounds that may inhibit the respiratory burst of neutrophils [17,18,19,20,21]. Furthermore, nutritional requirements may lead to the release of fatty acids that can contribute to skin irritation [10]. However, how these properties contribute to virulence has not been studies in depth in different infection models.

*Malassezia* yeasts are prominent members of the skin mycobiota and are considered to be commensals. Understanding the transition from a commensal microorganism to a pathogen in skin and systemic diseases is a major aim in current research. Besides, identifying the predisposing and risk factors of the host that contribute to this transition and also the response of the yeasts to these changing conditions can be studied and may help in the development of new therapeutic alternatives. The aforementioned goals require the implementation of infection models in which the virulence properties of *Malassezia* spp. can be studied. Depending on the formulation of the research question, different types of infection models might be used. However, to unravel host–microbe interactions, it is necessary to study infections in more than one model since each model system has its own properties and limitations. This review aims to show different infection models that have been used in the study of the *Malassezia* genus to understand the virulence properties of these yeasts and we will describe novel alternative models that are gaining importance in this field.

**Table 1 jof-06-00155-t001:** Diseases associated with *Malassezia* spp.

Disease	Clinical Findings	Species Involved	Most Commonly Affected Population	References
Pityriasis versicolor (PV)	Macules on the trunk and arms; the skin lesions are hypopigmented and hyperpigmented	*Malassezia globosa*, *Malassezia sympodialis*, and *Malassezia furfur*	Young adults and rarely children and older adults	[3,22,23,24,25,26,27]
Dandruff/seborrheic dermatitis (D/SD)	Flaking and erythema in sebum-rich areas like the scalp, nostrils, chest, and eyebrows	*M. globosa*, *Malassezia restricta*, *M. furfur*, and *Malassezia obtusa*	Elders, infants, children in puberty and HIV patients	[3,25,27,28,29,30,31,32,33]
Atopic dermatitis (AD)	Chronic inflammatory illness with pruritic eczematous lesions. *Malassezia* has been proposed to act as an exacerbator	*M sympodialis*, *M. globosa*, *M. furfur*, *M. restricta*, *Malassezia japonica*, *Malassezia yamatoensis*, and *M. slooffiae*	Adults with genetic and environmental predisposing factors	[3,27,34,35,36,37,38]
Folliculitis	Small dome-shaped papules localized around follicular areas, mainly in the back, chest, and shoulders. The papules can evolve into pustules	*M. globosa*, *M. restricta*, *M. sympodialis*, *M. furfur*, and *M. pachydermatis*	Teenagers and young adult males	[3,25,26,31,32]
Psoriasis	Chronic skin disease, characterized by hyperproliferation and hyperkeratinization of the epidermis. *Malassezia* may augment inflammation and the severity of the disorder	*M. globosa*, *M. furfur*, *M. sympodialis*, *M. restricta*, and *M. slooffiae*	Patients with psoriasis, mainly on the scalps of young adults	[3,27,37,38,39,40,41,42]
Crohn´s disease	Inflammatory bowel disease characterized by altered immune response to intestinal microbiota. *Malassezia* yeasts in the gut may increase the severity of the disease	*M. restricta*	Crohn´s disease patients carrying the CARD9^S12N^ risk allele	[4,43]
Parkinson’s disease	Neurodegenerative disease. Seborrheic dermatitis has been strongly associated with this disease	*M. globosa*, *M. restricta*, *M. furfur*, and *M. obtusa*	Elders. Risk increases after a seborrheic dermatitis diagnosis	[5]
Pancreatic ductal adenocarcinoma	Carcinoma due to fungal dysbiosis	*M. globosa*	Individuals with oncogenic *Kras* that induces inflammation, resulting in fungal dysbiosis	[6]
Invasive infections	Fungemia, endocarditis, bronchopneumonia, respiratory distress, splenic lesions, etc.	*M. furfur*, *M. pachydermatis*, *M. sympodialis*, and *M. restricta*	Low-weight neonates and immunocompromised patients	[3,7,8,26,44,45,46,47,48,49,50,51]

## 2. Infection Models as a Way to Understand Host–Microbe Interactions

Little is known about the virulence properties and infection mechanisms of *Malassezia* spp., and the implementation of infection models may allow for the evaluation of the interaction of these yeasts with hosts, the virulence of different species or strains of a specific species, and antifungal activity. There are different types of suitable models in which virulence and infection can be studied, but it is critical to realize that the results obtained in each model provide partial answers, as was mentioned before. It is therefore important to study virulence properties in different in vitro and in vivo models and the results obtained can provide complementary answers [52,53,54,55].

One of the infection models that may help to unravel host–microbe interactions is in vitro models, which have been used since the 1960s [55]. In vitro models are generally easier to handle, the majority of factors can be controlled, the evaluation of drug activity is more accurate, and, in some cases, they are cheaper than using animal models. These models can also be cataloged as ex vivo models [52,56,57], like cultured cells, removed organs, and skin equivalents or dermis equivalents [54,58,59]. As good as the in vitro models are, they do not fully reproduce the host–microbe interactions that occur, for example, on the skin.

Contrary to in vitro models, the in vivo models mimic the complexity of the host response better [53,54,57,60]. These are rather diverse and can vary from mammalian models to insect models. Mammalian models are phylogenetically the closest to human beings and, generally, are regarded as more accurately reproducing the host–microbe interaction, known as fidelity [52,54,57]. Additionally, many of these models are well characterized, allowing for genetic modifications to reach a desirable condition. The drawbacks of these models are the high cost of feeding and maintenance, the limited number of individuals, the ethical implications, and the need for trained personnel to handle the animals [54]. These drawbacks can be solved by the implementation of alternative animal models, like invertebrates.

Invertebrate animal models have recently gained importance in fungal research since studies have shown that the microbial virulence factors involved in infections in mammals are the same as those involved in invertebrate infections [53]. In fact, it seems that different aspects of the innate immune response in vertebrates and invertebrates are shared and represent a conserved trait, which means that human pathogens, at least in part, interact similarly with both immune systems [53,61]. The innate immune responses in invertebrate models are comparable to, for example, the human immune response to fungi via Toll-like receptors, which were originally discovered in *Drosophila melanogaster* [62], a model system already used with *Malassezia* [63], and also present in *Caenorhabditis elegans* [64]. Besides, the well-developed phagocytic system in lepidopterous and coleopterous larvae parallels the process of phagocytosis in mammalian systems [53,60,65,66,67,68,69].

### 2.1. In Vitro Models of Host-Microbe Interaction

In fungal infection research, the in vitro (ex vivo) models have been used to elucidate the mechanisms of interaction between fungi and their hosts. Indeed, an ex vivo model allows for the identification of the specific host tissue response to a pathogen, but it does not depict the whole host response [52,54,57]. An example of this is the implementation of keratinocytes to evaluate the response of these cells to skin-related fungal infections. *Trichophyton rubrum* was shown to induce the production of skin-derived antimicrobial peptides (AMPs) in primary keratinocytes, which may help the host to control dermatophyte infection [70]. Similarly, this model has been used as an infection model for *Candida albicans,* identifying the induction of proinflammatory cytokine production [71] and proteins involved in fungal adhesion to keratinocytes and interaction with the host [72].

The co-culturing of human keratinocytes with *M. furfur* yeasts was used to evaluate the activity of the cecropin A(1-8)–magainin 2(1-12) hybrid peptide analog P5 (an AMP). This research showed that this therapeutic alternative can indeed inhibit *M. furfur* growth without causing damage to keratinocytes. Moreover, AMPs can also modulate the inflammatory response of keratinocytes; this opens up the opportunity to evaluate new therapeutic alternatives in co-cultures of *Malassezia* and human keratinocytes, evaluating not just the drug effect on the pathogen but also the drug effect on and via the host [73]. Other studies have reported that *Malassezia* can induce or repress the production of cytokines in keratinocytes. The level of production depends on the species [74,75,76], the growth phase, and the hydrophobicity [77], and is affected by keratinocyte invasion and the survival of the pathogen inside the host cells [78]. In addition, it has been observed that *M. pachydermatis,* a zoophilic species, can invade human keratinocytes (12.1%) [79] and induce a strong inflammatory response during the first 24 h after coincubation [79,80]. In contrast, *M. furfur* has shown a lower induction of the inflammatory response, something that may be related to the avoidance of phagocytosis [78]. Interestingly, the presence of a capsule-like lipid layer may reduce the pro-inflammatory cytokine production in keratinocytes, as a way to evade the immune response [81].

In addition, the role of some factors that are excreted by species of *Malassezia* can be elucidated through in vitro model experiments. For example, the extracellular nanovesicles of *M. sympodialis* were co-cultured with keratinocytes and monocytes, demonstrating for the first time that these small structures are phagocytized by keratinocytes and monocytes [34]. Later, it was demonstrated that these nanovesicles play an important role in activating the keratinocytes as part of the cutaneous defense against *Malassezia* [82]. Furthermore, *M. furfur* has also been shown to secrete extracellular vesicles that can induce the production of pro-inflammatory cytokines in human keratinocytes. Additionally, similar to what was reported in *M. sympodialis*, the vesicles secreted by *M. furfur* are phagocytized by keratinocytes [83].

Another in vitro model is the skin equivalent (SE) generated from the isolation and cultivation of fibroblasts and keratinocytes. This system allowed the growth of an inoculum of 1 × 10^2^ CFU/mL of *M. furfur*, which grew to 1 × 10^4^ CFU/mL, which could mean that SE may produce and release the nutrients necessary for *Malassezia* to grow on this surface. This model appeared to mimic the lipid production by the host since the culturing media did not contain these lipids [58], but care must be taken that growth is not due to lipids associated with yeast cells and/or carried over from lipid-rich media used for pre-culturing. Similar to SE, there are other models that may allow for the understanding of the host response to *Malassezia*. For example, the reconstructed human epidermis (RHE) offers the opportunity to follow the progress of the infection over time and measure products of the immune response at every time point. In this case, it has been reported that *M. furfur* and *M. sympodialis* suppressed the inflammatory response after 48 h, thereby evading the host immune system. Additionally, this model showed again that the keratinocyte response pattern depends on the *Malassezia* species used, indicating that virulence properties and mechanisms of pathogenesis differ between them [59].

### 2.2. In Vivo Models of Host–Pathogen Interactions

#### 2.2.1. Mammalian Models of Host–Pathogen Interactions

Mammalian in vivo fungal infection models include mice, rats, guinea pigs, dogs, and rabbits [54,57,84,85]. In fungi, these models have allowed for the elucidation of the role of virulence factors, like the formation of biofilms of *Candida albicans* using rabbits and rats as infection models [57]. Immunosuppressed rats and mice have also been used as animal models to study invasive rhinosinusitis caused by *Aspergillus fumigatus* [86] and drug evaluation in pulmonary aspergillosis [87]. Furthermore, mice models were used to establish keratitis infections with fluorescently labeled *Fusarium solani*, allowing for the in vivo observation of the pathogens during infection [88].

For *Malassezia*, the implementation of a host model has been difficult due to the weak virulence of the species of this genus. The first attempts to develop a suitable model for *Malassezia* failed because an infection could not be established in the animal model or the infection was resolved in a short time period. In 1940, Moore et al. inoculated *M. furfur* directly on the intact skin of rabbits, guinea pigs, rats, and mice, which resulted in no establishment of the infection unless they were infected by intracutaneous or intratesticular inoculation [85]. The evaluation of the efficacy of antifungal treatments against *M. furfur* in guinea pigs was possible but required daily direct inoculation on intact skin for one week, which caused skin alteration that resembled SD [89]. Similar results were observed for *M. restricta* inoculated directly on the skin surface of guinea pigs; wherein severe inflammation was observed after repeated inoculation every 24 h over 7 days. The skin inflammation lasted for 52 days and resembled SD. Furthermore, in this study, it was possible to evaluate the antifungal activity of ketoconazole and luliconazole, showing that the efficacy of ketoconazole is correlated with clinical findings using ketoconazole as an antifungal agent against *Malassezia* spp. For luliconazole, it was observed that this antifungal significantly reduced *M. restricta* rDNA copies and skin lesions. Taken together, these results demonstrated the suitability of the guinea pig, not just as an infection model, but also to evaluate antifungal activity [90].

Dogs were also used to model external otitis caused by *M. pachydermatis*; this was done through the instillation of *M. pachydermatis* inoculum into the external ear canal. The aim of this inoculation was to evaluate the activity of antifungals on external otitis development. Dogs were examined daily and a microscopical examination of ear exudate was done. The results showed the development of external otitis with an erythematous ear canal and exudate production. Additionally, abundant *M. pachydermatis* yeasts were recovered in cultures from the samples [91].

A couple of experiments have been conducted in rabbits, inoculated directly on the surface of the skin with or without occlusion with a plastic film over the inoculated area to favor colonization; this led to the occurrence of lesions on the skin and the appearance of mycelial structures in histological studies. Again, it was observed that, as soon as inoculation with yeast cells was discontinued, spontaneous healing occurred. It was, furthermore, evident that infection only occurred when occlusion was employed [92,93,94]. The presence of *Malassezia* in healthy skin and the high development of seborrheic dermatitis (SD) infections in acquired immune deficiency syndrome (AIDS) patients led to the belief that these yeasts were opportunistic [95,96]. In that way, new strategies to mimic the conditions of susceptible hosts were implemented. In 2004, Oble et al. developed a novel transgenic T-cell model in mice, in which spontaneous SD-like disease developed. Using anti-fungal staining, ovoid structures in primary lesions were observed. Furthermore, antifungal treatment resulted in the reversion of clinical symptoms. Although fungi were not isolated and characterized from the lesions, overgrowth by *Malassezia* spp. seems plausible, suggesting that infections only occur under conditions of severe immunological impairment [97].

Starting from this point, it is clear that animal models must have some kind of predisposition or repetitive exposure to successfully develop fungal infection with *Malassezia*. Yamasaki el al. developed a new deficient Mincle mouse model for *Malassezia*. Mincle, also known as Clec4e, is a PRR that recognizes the PAMP mannosyl-fatty acid in *Malassezia*. With the Mincle-deficient mice, it was demonstrated that the recognition of this PAMP induced the release of the cytokines Il-6 and TNF in the host, similarly to that observed in *Malassezia*-induced lesions in humans [98]. Another way of causing immunosuppression in animal models is through the employment of chemical substances like hydrocortisone and cyclophosphamide, which results in a different type of immunosuppression. The latter results in neutropenic animals [99].

Predisposing factors include skin barrier disruption. In 2019, Sparber et al. demonstrated that epicutaneous infection by *Malassezia* spp. can be established by disrupting skin integrity using an adhesive tape on the dorsal skin of the ear of a mouse. This study showed that *Malassezia* induces the release of IL-17, which stimulates tissue inflammation, agreeing with findings in atopic dermatitis [15]. Recently, a new model for experimental psoriasis has been proposed, wherein imiquimod was employed to induce psoriasis-like dermatitis in a murine model. The results from this study support the idea that the presence of *Malassezia* on the skin may augment the skin disorder or induce it [41]. As can be seen with respect to mammal models, new in vivo alternatives have now been proposed that facilitate *Malassezia* infection in animals.

#### 2.2.2. In Vivo Alternative Models of Host–Microbe Interactions

In general, in vitro studies allow for the finding of patterns that require subsequent testing and validation in in vivo infection systems; ethical considerations have especially pushed the development of new model systems. With respect to animal treatment, Russell and Burch proposed the 3Rs strategy (replacement, reduction, and refinement). This strategy leads to reducing the use of mammals and the replacement of these with alternative models; like computer, in vitro, alternative vertebrate (*Danio rerio*) [100], and invertebrate models [101]. In general, invertebrate alternatives used to model fungal infections like amoeboid models [53,102], *Caenorhabditis elegans* [103,104,105], *Drosophila melanogaster* [63,106,107], *Tenebrio molitor* [108], *Bombyx mori* [60], and *Galleria mellonella* [65,67,109,110] (Figure 1 and Table 1) have gained importance, amongst others, as these present an innate immune response similar to that found in mammals. Furthermore, microbial virulence factors play similar roles in mammals and invertebrate systems [53,106,111]. The results obtained with these models correlated with results obtained in mammalian models, validating the invertebrates as infection models [106,111,112,113,114,115,116]. Furthermore, the attractive features of these models include the low cost of feeding and the higher number of organisms able to be stored in a small space and used in a single experiment [60].

**Table 2 jof-06-00155-t002:** In vitro and in vivo models available for *Malassezia* spp. infection studies.

Infection Model	Cost	Inoculation	Advantages	Disavantages	References
Keratinocyte culture	High	-Co-culture	-Controlled conditions -Just one type of cell	-It does not represent the complex interactions with the host	[52,54,74,75,76,77,78,79,80,81,83]
Murine model	High	-Oral gavage -Inoculation through the tail vein -Inhalation and intranasal administration -Direct inoculation -Ocular -Intracranial -Intraperitoneal	-Well-defined inoculation routes -Immune response is similar to a human’s, with innate and adaptative immune response -Mimics human infection and disease -Annotated genome -Available mutants	-Ethical issues -Bigger space for storage -Longer generation time -Trained personnel to handle the models -Immune suppression required	[57,60,117,118]
Amoeboid model (*Acanthamoeba castellani*)	Low	-Co-incubation	-Controlled conditions -Inoculum quantification -Available mutants -Incubation at 37 °C -Phagocytosis assays -Short life cycle -Annotated genome	-Undesired mutation and loss of phagocytic abilities in long-cultured strains	[53,54,102,119,120]
Zebrafish larvae (*Danio rerio*)	Low	-Microinjection into the caudal vein, notochord, duct of Cuvier, hindbrain ventricle, eye, peritoneal cavity, or muscle - exposure by immersion	-Short generation time -Annotated genome sequence -Available mutants -Transparency -High-throughput screening -Innate immune response similar to that of humans	-In larval stage, there is no adaptative immune response -Ethical issues in some countries -Difficult to handle	[100,118,121,122,123,124,125,126]
*Caenorhabditis elegans*	Low	-Exposure of larvae by immersion (feeding and contact with the cuticle)	-Short generation time -Small size -Easy to grow -Annotated genome sequence -Available mutants -Innate immune response similar to that of humans -Results correlated with results from mammals	-There is no adaptative immune response -Difficult to inoculate and quantify the inoculum	[87,127]
Silkworm *(Bombyx mori*)	Low	-Microinjection into the haemocoel -Oral (puncture)	-Inoculum quantification -High inoculum volume -Results correlated with results from mammals -Innate immune response similar to that of humans	-No adaptative immune response	[60,118,128]
*Drosophila melanogaster*	Low	-Puncture in the dorsal side of the thorax	-Annotated genome sequence -Available mutants -Innate immune response similar to that of humans	-No adaptative immune response -Difficult to inoculate and quantify the inoculum	[63,106,129]
*Galleria mellonella*	Low	Microinjection directly to the haemocoel -Topical -Oral	-Inoculum quantification -Wide range of temperatures -Innate immune response similar to that of humans -Available immune response transcriptome -Results correlated with results from mammals	-No adaptative immune response	[67,109,118,119,130,131,132,133]

In the field of *Malassezia* research, hardly any work has been published with alternative in vivo models and the implementation of invertebrates as model systems is very recent. In 2018, Brilhante et al. implemented for the first time the *C. elegans* larva as an infection model for *M. pachydermatis*. In this study, *C. elegans* larvae were exposed to *M. pachydermatis* by placing the larvae in plates containing the yeasts for a period of two hours at 25 °C. The viability of the nematodes was evaluated every 24 h and the results showed that, after 96 h, the nematodes exposed to the yeast had significantly higher mortality (ranging from 48% to 95%) than the control nematodes [134]. After that, in the same year, Silva et al. also evaluated the virulence of *M. furfur*, *M. sympodialis*, and *M. yamatoensis* under different growth conditions. The implementation of *C. elegans* larvae resulted in the identification of different virulence patterns depending on the lipid supplementation of the pre-culture medium. The co-culture of larvae with *Malassezia* spp. grown in media that was not supplemented with lipids resulted in lower larval survival. In the same study, a second model was implemented. *T. molitor* larvae were inoculated with a yeast suspension, and the larvae were shown, as in the case of *C. elegans*, to have higher survival when inoculated with *M. furfur* grown in a lipid-supplemented medium [135]. These two models allowed them to assess the virulence of three species of *Malassezia* under different growth conditions. However, more research needs to be done to understand this phenomenon.

In addition to *T. molitor* larvae, other insects have been implemented recently as an infection model for *Malassezia*. That is the case for *D. melanogaster*. Wild type (*WT*) and *Toll*-deficient adult flies were inoculated with five different inoculum concentrations of *M. pachydermatis*. The results showed that *WT* flies were resistant to the infection and that *Toll*-deficient flies inoculated with the highest inoculum concentrations showed a significantly reduced survival as compared to the control. These findings were corroborated with a decrease in fungal burden in *WT* flies and an absence of yeasts in histological investigations, contrasting to what was observed in the *Toll*-deficient flies [63]. These results demonstrated the opportunistic character of *M. pachydermatis* and showed the potential of the use of immune-deficient mutant flies to study the pathogenesis of *Malassezia*.

The *G. mellonella* larva was first used as a fungal infection model in 2000. In that study, the virulence of *C. albicans* was evaluated and compared with the effect of inoculating the larvae with *Saccharomyces cerevisiae.* The results showed that inoculating the larvae with the former had a lethal effect. In contrast, *S. cerevisiae* was not shown to be pathogenic. Additionally, it was found that clinical isolates of *C. albicans* were more virulent as compared to reference strains (ATCC 10231, ATCC 44990, and MEN). These results correlated with findings in mammalian models [112]. After this, *G. mellonella* has been widely implemented as a fungal infection model to evaluate virulence [111,116,119,131,136,137,138], virulence patterns related to biofilm formation [133], co-infections [113], pathogen morphogenesis [115], complex host responses [114,139,140,141,142], and antifungal susceptibility [110,143,144,145,146] at 37 °C, which is an advantage of this lepidopteran, as it can be incubated at human physiological temperatures. The results of most of these studies have been shown to correlate with results obtained in mammalian models and also in humans. Indeed, the efficacy of antifungals tested in *G. mellonella* larvae against *Cryptococcus* spp. [110] and *Candida* spp. [146] have been shown to correlate with results in the murine model, *C. elegans* larvae, and in vitro models. Even though these results are interesting, there is a need to better understand this insect. At present, there is available information related to the immune response transcriptome [147] and the miRNAs involved in the regulation of the immune response [148] that can help to evaluate the host response to a specific pathogen. All of this together makes this insect a promising tool to elucidate the complex host–microbe interactions of *Malassezia*.

*G. mellonella* has been standardized as an infection model for *M. furfur* CBS 1878 and *M. pachydermatis* CBS 1879, two isolates from skin lesions. The inoculation of larvae with these two species showed that larval survival depended on the inoculum concentration (higher inoculum concentration led to lower survival, compared to lower inoculum concentration). Additionally, a lower virulence was observed for *M. furfur* as compared to *M. pachydermatis* at 33 °C and 37 °C. This was evident by a decrease in larval survival, higher fungal burden, histological examination with a higher presence of hemocyte aggregates with melanin deposition, and a higher larval melanization, especially in larvae that were inoculated with *M. pachydermatis* and incubated at 37 °C. The higher virulence of *M. pachydermatis* was attributed to a high phospholipase activity and a high capacity of *M. pachydermatis* to form biofilms [149]. However, further studies are required to confirm these hypotheses. These results show that the *G. mellonella* larva is a suitable model and very useful to identify differences in the virulence between species or strains.

## 3. Conclusions

The use of both in vitro and in vivo models is important in unraveling the interactions between microbes and hosts, and a variety of models are indeed available (Figure 1 and Table 2). In vivo models clearly allow for the direct comparison of virulence and studies of pathogenic developments in the host, and are, in that respect, most attractive. Alternative models that can replace mammalian models on ethical grounds are favored to reduce the number of animals used in research. However, it is important to keep in mind that alternative models do not completely replace mammalian models. In short, insect larvae like the *G. mellonella* larva have been proven to be reliable models and produce results similar to those reported in the murine models, making it an interesting tool to decipher aspects of the host–microbe interactions of *Malassezia.*

## Figures and Tables

**Figure 1 jof-06-00155-f001:**
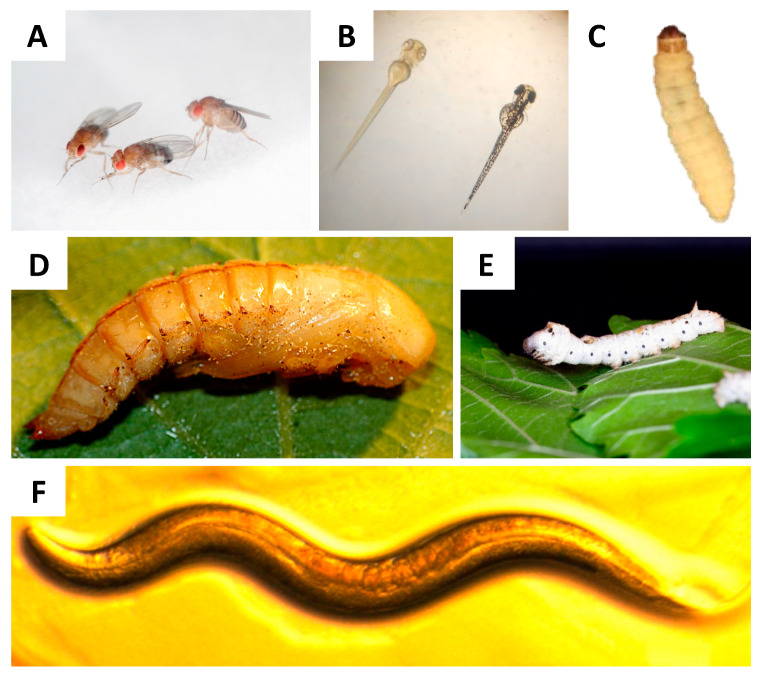
Alternative in vivo models for host–microbe interaction studies. (**A**) Adult *D. melanogaster* fly, whose size is approximately 3 mm. Original photograph by Flickr user NASA’s Marshall Space Flight Center, CC BY-SA 2.0 license. (**B**) *Danio rerio* larval size can range from 3.5 mm to 11 mm and, as can be seen, larvae are transparent, this facilitates monitoring the progress of the infection. Original photograph by Flickr user MichianaSTEM, CC BY-SA 2.0 license. (**C**) *G. mellonella* larval size ranges from 2 cm to 3 cm and its weight ranges between 200 mg and 300 mg, making it easy to manipulate and inoculate. (**D**) *T. molitor* pupae, easy to breed and the size at the 2nd instar is similar to that of *G. mellonella*. Original photograph by Flickr user Edithvale-Australia Insects and Spiders, CC BY-SA 2.0 license. (**E**) *B. mori* larvae, these larvae are large and their weight is in the range of 900 mg to 1000 mg. Original photograph by Flickr user Gianluigi Bertin, CC BY-SA 2.0 license. (**F**) *C. elegans* nematodes, which grow to 1 mm. Original photograph by Flickr user NIH Image Gallery, CC BY-SA 2.0 license.

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
