# Peer review of "In Vitro or In Vivo Models, the Next Frontier for Unraveling Interactions between *Malassezia* spp. and Hosts. How Much Do We Know?"

_jof, 2020, doi:10.3390/jof6030155_

Round 1

Reviewer 1 Report

I attached the review with my comments as sticky notes

Author Response

Dear reviewer,

We appreciate your comments and suggestions, we are sure these will highly contribute to the improvement of our manuscript. Here we show a list of the corrections and answers to your question.

1. No relationship was found between MIC values and outcome among the models?

Yes, results in Candida krusei and Candida albicans have shown to correlated with results in the murine model and C. elegans larvae (Scorzoni et al., 2013). Other studies have also reported similar results, wherein the efficacy of antifungals shows a correlation between different models and also with the in-vitro model (Jemel et al., 2020).

Scorzoni, L.; de Lucas, M.P.; Mesa-Arango, A.C.; Fusco-Almeida, A.M.; Lozano, E.; Cuenca-Estrella, M.; Mendes-Giannini, M.J.; Zaragoza, O. Antifungal Efficacy during Candida krusei Infection in Non-Conventional Models Correlates with the Yeast In Vitro Susceptibility Profile. PLoS One 2013, 8, 1–13.

Jemel, S.; Guillot, J.; Kallel, K.; Botterel, F.; Dannaoui, E. Galleria mellonella for the evaluation of antifungal efficacy against medically important fungi, a narrative review. Microorganisms 2020, 8, 1–20.

2. Malassezia is more like a skin pathogen

Malassezia genus is usually found in healthy individuals. Indeed, yeasts of this genus have been found to be predominant eukaryotic microorganisms in healthy human and animal skin. Under specific conditions these yeasts become pathogens, but normally, they are found as part of the skin micobiota (Wu et al., 2015).

Wu, G.; Zhao, H.; Li, C.; Rajapakse, M.P.; Wong, W.C.; Xu, J.; Saunders, C.W.; Reeder, N.L.; Reilman, R.A.; Scheynius, A.; et al. Genus-Wide Comparative Genomics of Malassezia Delineates Its Phylogeny, Physiology, and Niche Adaptation on Human Skin. PLoS Genet. 2015, 11, 1–26.

3. You Highlighted the sentence corresponding to the goal of this review “This review aims to show the importance of different infection models in the study of the Malassezia genus to understand the virulence properties of these yeasts, and we will describe novel alternative models that are gaining importance in this field”

We are not sure why you highlighted it, but, in order to be more coherent with the review construction we made a correction in this goal as following:

“This review aims to show different infection models that have been used in the study of the Malassezia genus to understand the virulence properties of these yeasts, besides we will describe novel alternative models that are gaining importance in this field”

4. You suggested that psoriasis is seen frequently among older patients both women and men.

We made the correction.

5. The reference 49 that corresponds to the following citation “Mylonakis, E.; Casadevall, A.; Ausubel, F.M. Exploiting amoeboid and non-vertebrate animal model systems to study the virulence of human pathogenic fungi. PLoS Pathog. 2007, 3, e101, doi:10.1371/journal.ppat.0030101” was highlighted. We are not sure about the reason, may you explain us why it was highlighted? However, as you can checked in this paper, the authors refer the utility to those models, as they mention: “A recurrent finding in recent studies of fungal virulence factors is that many of the same pathogenesis traits are required for virulence in both mammals and non-vertebrate hosts”.

6. In line 116 you highlighted “cecropin”, we are not sure why, but we are referring here to the reference 69: Ryu, S.; Choi, S.Y.; Acharya, S.; Chun, Y.J.; Gurley, C.; Park, Y.; Armstrong, C.A.; Song, P.I.; 518Kim, B.J. Antimicrobial and anti-inflammatory effects of cecropin A(1-8)-magainin2(1-12) hybrid peptide analog P5 against Malassezia furfur  infection in human keratinocytes. J. Invest. Dermatol. 2011, 131, 1677–1683, doi: 10.1038/jid.2011.112.

We change it as follow: “Cecropin A(1-8)–Magainin 2(1-12)” you can find it in line 120

7. You made a grammar suggestion in line 130, to add “by species of Malassezia

We corrected this. It can be found in line 136.

8. You suggested to explain first what the abbreviation Clec4e meant and to use then the abbreviation

We solved it as “Mincle, also known as Clec4e”. it can be found in line 209.

9. As you suggested we added the term larvae after elegans. It can be found in line 272.

10. You commented that mellonella larva has been used for C. tropicalis too, which is true. In fact, as was mentioned in the manuscript, G. mellonella have been implemented as an infection model for C. tropicalis to evaluate virulence and the efficiency of antifungals.

Since the manuscript is in preprint, we had a feedback from Dr. Martin Laurence. He suggested us four new references that we found very interesting, this were integrated into the review (references 38, 41, 42 and 43).

Sincerely, we hope this will clarify any doubts you have and meet your expectations

Our regards

Reviewer 2 Report

The authors are to be commended for their tremendous effort to compile a large number of test systems for various Malassezia strains. The result is a very useful source of references and models for researchers from different areas of natural sciences who intend to investigate host-Malassezia relations, immune response to Malessezia and maybe treatments. The non-vertebrate models are particularly promising as they are cheep, easy to maintain and available in great numbers; their disadvantage is the lack of development of adapted immunity.

Tjere are a number of typos - some of them listed below - and grammatical issues to be corrected.

Line 58: ... is a prominent member of ...

Line 60:  .. predisposing and risk factors ...

Line 142: .. this model appeared to mimic ...

Line 150: .. keratinocyte response pattern ..

Line 151 : ... species

Line 171:  .. during 7 days

Lines 172-173: antifungal activity of keto...

Line 195: Although fungi were ...

Table 2: "Just one type of cells

.. tail vein..

... there is no ...

Results correlated with results ...

Difficult to inoculate ..

Line 271: "... what was observed ...

Lines 276-277: "In contrast, S cerevisiae ..."

Line 280: ",,, model to evaluate ..."

Author Response

Dear reviewer,

We really appreciate your comments and suggestions, they are a very good feedback to our work and the corrections have been already made in the document. You can find the corrections in the manuscript since we are using track changes. Even thought, we are given you a list of the correction that were made and were you can find them.

1. In line 58, you suggested to change it to “Malassezia is a prominent member of” we changed to “Malassezia yeasts are prominent members of”. It can be found in line 58.

2. In line 60, you suggested to change it to “predisposing and risk factors”, we did it. It can be found in line 60.

3. In line 142, you suggested to change it to “this model appeared to mimic…”, we did it. It can be found in line 148.

4. In line 150, you suggested to change it to “keratinocyte response pattern …”, we did it. It can be found in line 156.

5. In line 151, you suggested to change it to “species …”, we did it. It can be found in line 157.

6. In line 171, you suggested to change it to “during 7 days …”, we did it. It can be found in line 177.

7. In lines 172-173, you suggested to change it to “antifungal activity of ketoconazole…”, we did it. It can be found in line 178-179.

8. In line 195, you suggested to change it to “Although fungi were …”, we did it. It can be found in line 204.

9. In the table 2, you suggested a few changes, which were... tail vein..., ... there is no ..., Results correlated with results ... and Difficult to inoculate. We made the right corrections in the table; the corrections are highlighted.

10. In line 271, you suggested to change it to “what was observed …”, we corrected it. It can be found in line 293.

11. In lines 276-277, you suggested to change it to “In contrast, cerevisiae …”, we changed it. It can be found in line 300.

12. In line 280, you suggested to change it to “… model to evaluate …”, we corrected it. It can be found in line 303.

Since the manuscript is in preprint, we had a feedback from Dr Martin Laurence. He suggested us four new references that we found very interesting, this were integrated into the review (references 38, 41, 42 and 43).

We hope this will meet your expectations for the suggested minor revisions

Our regards,